# Attachment to Meat and Willingness Towards Cultured Alternatives Among Consumers: A Cross-Sectional Study in the UAE

**DOI:** 10.3390/nu17010028

**Published:** 2024-12-25

**Authors:** Sharfa Khaleel, Tareq Osaili, Dana N. Abdelrahim, Falak Zeb, Farah Naja, Hadia Radwan, MoezAlIslam E. Faris, Hayder Hasan, Leila Cheikh Ismail, Reyad S. Obaid, Mona Hashim, Wael Ahamd Bani Odeh, Khalid Abdulla Mohd, Hajer Jassim Al Ali, Richard A. Holley

**Affiliations:** 1Research Institute of Medical and Health Sciences, University of Sharjah, Sharjah P.O. Box 27272, United Arab Emirates; sharfakhaleel@gmail.com (S.K.); danaabdelrahim@gmail.com (D.N.A.); falak106@gmail.com (F.Z.); fnaja@sharjah.ac.ae (F.N.); hradwan@sharjah.ac.ae (H.R.); haidarah@sharjah.ac.ae (H.H.); lcheikhismail@sharjah.ac.ae (L.C.I.); robaid@sharjah.ac.ae (R.S.O.); mhashim@sharjah.ac.ae (M.H.); 2Department of Clinical Nutrition and Dietetics, College of Health Sciences, University of Sharjah, Sharjah P.O. Box 27272, United Arab Emirates; 3Department of Nutrition and Food Technology, Faculty of Agriculture, Jordan University of Science and Technology, Irbid 22110, Jordan; 4Nutrition and Food Sciences Department, Faculty of Agriculture and Food Sciences, American University of Beirut, Riad El-Solh, Beirut P.O. Box 11-0236, Lebanon; 5Department of Clinical Nutrition and Dietetics, Faculty of Allied Medical Sciences, Applied Science Private University, Amman 11931, Jordan; moezalislam@gmail.com; 6Department of Women’s and Reproductive Health, University of Oxford, Oxford OX3 9DU, UK; 7Food Studies and Policies Section, Food Safety Department, Dubai Municipality, Dubai P.O. Box 67, United Arab Emirates; wmodeh@dm.gov.ae (W.A.B.O.); halali@dm.gov.ae (H.J.A.A.); 8Food Inspection Section, Food Safety Department, Dubai Municipality, Dubai P.O. Box 67, United Arab Emirates; kamohd@dm.gov.ae; 9Department of Food and Human Nutritional Sciences, Faculty of Agriculture and Food Sciences, University of Manitoba, Winnipeg, MB R3T 2N2, Canada; rick.holley@umanitoba.ca

**Keywords:** meat consumption, cultured meat alternatives, consumer attitudes, food preferences, sustainable eating

## Abstract

**Background/Objectives**: The escalating global demand for meat, as a sequela of population growth, has led to unsustainable livestock production, resulting in a host of environmental and food security concerns. Various strategies have been explored to mitigate these issues, including the introduction of a novel food product, cultured meat. Cultured meat is not yet commercially available, yet public perceptions are already taking shape. To better understand the factors influencing its adoption by consumers, a cross-sectional, web-based study was conducted to examine consumer attitudes toward conventional meat and cultured meat among adults in the United Arab Emirates (UAE). **Methods**: The survey was conducted between December 2023 and March 2024 and used a convenience snowball sampling method. The questionnaire focused on current meat consumption patterns, meat attachment, and willingness to consume cultured meat. Sociodemographic data, including age, sex, education, and self-reported weight and height, were also collected. **Results**: Results showed that the vast majority (86%) of participants consumed all types of meats, while more than half (59.3%) were unfamiliar with the term “cultured meat”. Despite this unfamiliarity, about one-third (35%) were somewhat willing to try cultured meat, though more than two-thirds (69%) were reluctant to replace conventional meat with cultured meat in their diet. Male participants and those with higher BMIs showed a significantly stronger attachment to conventional meat. Willingness to consume cultured meat was notably higher among participants aged less than 30 years, those having less formal education, and those who are Arabic. **Conclusions**: These findings suggest that while interest in cultured meat exists, significant barriers remain, particularly regarding consumer education and cultural acceptance.

## 1. Introduction

Worldwide, meat production comprising poultry, fish, and red meat has emerged as the primary contributor to fulfill the protein requirements of an expanding population, accounting for nearly one-third (33%) of global protein consumption [1]. The Food and Agriculture Organization (FAO) predicts that meat demand will increase by 73% to meet the needs of more than 10 billion people worldwide by 2050 [2,3]. It is estimated that conventional meat production employs around 83% of global farmland and may partly account for up to 70% of the planet’s deforestation whilst producing between 12% and 18% of the total greenhouse gas emissions [4,5,6,7]. Livestock production conducted to an unsustainable degree contributes to water depletion, accelerated climate change, and reduced biodiversity [8]. Research indicates that decreasing meat intake in both developed and transitioning nations, where consumption levels are either high or growing rapidly, is essential to alleviate additional pressure on the earth’s life support systems and ensure global sustainability [9,10].

A convergence of factors, including environmental sustainability, animal welfare, and evolving consumer preferences, has stimulated interest in alternative sources of protein [11]. Strategies for curtailing meat consumption have been explored, one option being cultured meat, sometimes termed cultivated meat or lab-grown meat. Cultured meat uses a novel technology to produce cell-based, lab-grown, and slaughter-free meat. The process involves muscle cells extracted from a live donor animal and grown in a culture medium in a laboratory setting. Controlled temperature, nutrients, and oxygen facilitate the multiplication and development of muscle fibers [12]. The primary benefits of cultured meat, as highlighted in numerous studies, are often rooted in concerns for animal welfare and the potential to significantly lower greenhouse gas emissions, approximately 78–96% of its original levels [13]. However, some of these aspects are still under discussion or are a matter of controversy. Moreover, it is proposed that introducing cultured meat to the market may address ethical concerns associated with large-scale factory farm operations [14]. Cultured meat represents a unique opportunity for individuals to consume real animal meat having conventional meat’s familiar organoleptic properties, including sensory, textural, and nutritional characteristics, without the need for animal slaughter [15].

Cell-based meat technology and subsequent cultured meat production are still in the nascent stages. As such, the introduction of alternative meat products to the market comes at a high cost [16]. Nevertheless, research is ongoing to address the technical challenges of advancing this technology, reducing associated costs, and expanding its reach to global markets as interest in cultured meat continues to grow. In 2020, Singapore approved the sale of a cultured meat product known as “Chicken Bites”, developed by the start-up company “Eat Just” [17], becoming the first country to adopt cultured meat products. Later, in 2022, 156 companies from 26 countries publicly announced their production of cultivated meat and seafood. In 2022, the U.S. Food and Drug Administration completed its premarket review and approved the sale of cultured chicken meat products from the UPSIDE Foods company [18]. Yet for all its potential environmental and public health promise, public acceptance of cultured meat remains ambiguous. Several studies have been conducted on the topic, and the results are mixed. Some studies have shown that consumers are unsure about accepting cultured meat [19,20,21], while others have found that there is a greater willingness to consume it [22,23]. Some studies have found that while certain respondents would eat cultured meat, most people still prefer conventional meat over cultured meat [24]. There is a need to understand current public perceptions of cultured meat.

A comprehensive review found that several factors determine consumer acceptance of cultured meat. These factors include meat consumption habits, public awareness, perceived naturalness, threshold for disgust, sensory appeal, and food safety perception [25]. Ethical and environmental concerns have motivated some consumers to pay extra for meat substitutes. However, this trend does not necessarily extend to cultured meat products. Furthermore, among several factors, Graça et al. demonstrated that meat attachment (MA) is an important construct when investigating consumers’ meat-eating habits and their willingness to accept meat substitutes, stating that an affective connection towards meat significantly influences consumer willingness to alter meat consumption habits [26]. Graca et al. created the MA scale, a relevant psychological construct representing meat intake and substitution [27]. Studies have demonstrated that a higher level of consumer attachment to meat correlates with a decreased willingness to reduce meat consumption and/or adopt meat substitutes [28,29]. Cultural and religious norms are significant drivers of consumer acceptance and behaviour [30]. While most research has focused on Europe [19,22,23], the United States [20], and Australia [25,31], exploring consumer attitudes in other regions, such as Arab countries, is essential. Meat production systems in many Arab countries are important sectors that underpin economic and social development at the national level [32]. Additionally, in Muslim-majority regions such as the Arab regions, halal certification is required for meat products, aligning with Islamic beliefs that govern permissible consumption in accordance with Islamic *Shariah* law [33]. The use of animals for cultured meat will be permissible (halal) as long as it adheres to the standards set by *Shariah* law [34]. The global halal food market is valued at USD 1.27 trillion and is projected to reach USD 1.67 trillion by 2025 [35]. In the UAE, where meat consumption is high [36] and interest in alternative protein sources is rising [37], studying public sentiment toward cultured meat is crucial. This interest is especially relevant following the UAE’s commitments at COP28 to implement greenhouse gas mitigation strategies to lower global warming to 1.5 °C [38]. This focus on sustainable practices highlights the importance of understanding public opinion on novel food solutions that align with global environmental agendas.

Thus, the present study seeks to investigate consumers’ perceptions of cultured meat in the UAE, exploring the factors that influence their willingness to try, consume, and purchase such meat. Furthermore, this study aims to explore the influence of MA, if any, in shaping these perceptions. This study is the first of its kind to delve into the attitudes of adults in the UAE towards the consumption of cultured meat. By examining the attitudes of consumers towards cultured meat, it is possible to compare these with other populations and, more broadly, understand factors that influence consumer willingness toward cultured meat consumption.

## 2. Materials and Methods

### 2.1. Study Design

A cross-sectional study was conducted to explore consumer meat attachment and their perspectives regarding cultured meat between December 2023 and March 2024 among adult consumers in the UAE. The survey was an online questionnaire with closed-ended questions in two languages: English and Arabic.

### 2.2. Study Population and Sample Size

Study participants were included in the study according to the following inclusion criteria: (1) adults aged 18 years or older and (2) currently residing in the UAE. A snowball convenience sampling technique was employed to achieve large-scale dissemination. The outcome of interest in this study is the consumer perspectives towards cultured meat. Sample size calculations showed that a minimum of 460 participants were required to estimate an outcome prevalence of 50% with a 5% margin of error and 95% confidence interval while accounting for a dropout attrition rate of 20%. The following calculation was used:$$N=Z^{2}\times P\times (1-P)/e^{2}$$
where *Z* = 1.96; *P* (estimated proportion of the population that presents the characteristic) = 0.5; *e* (margin of error) = 0.05; and *N* (sample size) = 384 participants, plus 20% (attrition rate) = 460 participants [39].

### 2.3. Ethical Considerations

Ethical approval was obtained from the Research Ethics Committee at the University of Sharjah, Sharjah, UAE, (Reference number: REC-23-09-04-01-F). The study was conducted following the ethical guidelines established by the Declaration of Helsinki in 1964. No identification information was collected in order to ensure the confidentiality and anonymity of participants involved. Before answering the questionnaire, an information dialog box explaining the study protocol and objective was provided on the first page. All participants provided electronic informed consent.

### 2.4. Study Questionnaire

The study employed a self-administered, online survey developed with Google Forms and included multiple components. The questions included in the online survey were adapted from Wilks et al. and Bryant et al. [20,40]. The questions were crafted using clear language to ensure the survey was concise and free of ambiguity. Screening questions ensured participants were over 18 and were residents in the UAE. The survey took about 7 min to complete, allowed participants to exit anytime, and stressed that participation was anonymous and voluntary. The survey link was shared on social media platforms like Facebook™, Reddit™, and WhatsApp™, encouraging further sharing.

### 2.5. Translation of the Questionnaire and Pilot Testing

The questionnaire was developed in English and later translated into Arabic. The Arabic version was back-translated by a professional translator to ensure parallel-form reliability. Two bilingual experts reviewed both the original and back-translated versions for consistency in meaning. Prior to launching the survey, the questionnaire was pilot tested on a sample of 20 participants, and the questions were assessed for clarity and cultural appropriateness. The data from the pilot testing were excluded from study analysis. Following the pilot testing, slight modifications were made to adapt the survey culturally and ensure its comprehension. As a result, a few questions were revised, as described below in the description of the data collection tool.

### 2.6. Description of the Data Collection Tools

The survey was divided into four sections.

#### 2.6.1. Section 1: Current Meat Consumption Pattern

The participants were asked to choose their current pattern of meat consumption (meat-eating: red meat, poultry, seafood; white meat only: poultry, seafood; pescatarian: only seafood; vegetarian: no food products made from meat, poultry, seafood, or animal by-products; or vegan: no food products that come from animals). Next was a multi-response question about the animal protein sources consumed most frequently in an average week (beef, mutton, poultry, fish, dairy products, eggs, or “I never eat any of the above”) and about the meals in which meat was consumed (breakfast, lunch, dinner, snacks, none).

#### 2.6.2. Section 2: Meat Attachment

An adapted version of the Meat Attachment Questionnaire (MAQ) was used in this study [27]. Meat attachment (MA) is a measure referring to a positive bond towards meat consumption. The MAQ is a 16-item validated questionnaire described using a 5-point Likert scale ranging from 1 (strongly disagree) to 5 (strongly agree). Multiple studies employ MAQ as a valid and reliable measure of the extent to which consumers feel a connection or preference for meat as part of their diet [29,41,42]. The present study included 14 questions retained from the original 16-item MAQ. Upon pilot testing, two statements, namely, “To eat meat is the unquestionable right of every person” and “According to our position in the food chain, we have the right to eat meat”, were removed to account for cultural sensitivity surrounding meat in the population of interest since the majority of the population in the UAE identify as Muslim [43]. In the MAQ questionnaire, the 14 items end up with a total score ranging from 14 to 70. Higher scores indicate a greater MA. In order to examine the study population in depth, the participants were classified based on their MA scoring. As there are no established guidelines for grouping, we employed the interquartile range (IQR) for classification. Consequently, the MAQ score was used to categorize the respondents into three groups based on the IQR: low MA (25%), medium MA (50%), and high MA (25%) [44].

#### 2.6.3. Section 3: Questions Assessing Willingness Toward Cultured Meat Consumption

This section explored the participant’s familiarity with the term “cultured meat” and willingness to consume cultured meat. One question assessed participant familiarity with the term “cultured meat”, and the rest of the questions tested their willingness to try, purchase, eat, or replace cultured meat for conventional meat and/or meat substitutes and, finally, if they were willing to pay a higher price for cultured meat. Next, a multi-response question explored which type of meat respondents would be willing to consume if it was produced using cultured methods. The participants were then asked to consider a hypothetical future where cultured meats were widely available. This questionnaire, which assesses willingness to accept cultured meat, was based on a 5-point Likert scale, ranging from 1 (not willing at all) to 5 (absolutely willing). The willingness section’s total score was between 5–25 points. The participants were then categorized based on the mean score into more willingness and less willingness.

#### 2.6.4. Section 4: Sociodemographic Questions

This section included questions regarding sociodemographic characteristics, including sex, age, highest level of education, employment status, marital status, nationality, resident emirate, and monthly household income. The weight and height were self-described also by the participants. Body mass index (BMI) was calculated by dividing the weight (kg) by the height (m^2^) and then classified following the WHO guidelines into four categories: underweight (<18.5 kg/m^2^), normal (18.5–24.9 kg/m^2^), overweight (25.0–29.9 kg/m^2^), and obese (>30.0 kg/m^2^) [45].

### 2.7. Statistical Analysis

Data analysis was conducted using IBM SPSS Version 29.0 (USA). The socio-demographic data were described as categorical variables using frequencies and percentages. Numerical variables, including age, weight, height, and body mass index, were described using means and standard deviations. The total score for MAQ and questions assessing willingness toward cultured meat consumption were obtained as a continuous variable and described as mean ± standard deviation, while the categories were represented as frequency and percentages. The comparison of the sociodemographic data among the MAQ and the willingness scores was done by independent sample *t*-test, ANOVA, and Chi-Square test. Simple linear regression was used to study the determinants of the total MAQ and willingness scores in the study population. The significance level for all the data was set at a *p* < 0.05.

## 3. Results

### 3.1. Sociodemographic Characteristics of the Study Population

Out of 618 participants who clicked the survey link, 577 consented to participate (response rate: 93.4%). The sociodemographic characteristics of the respondents are summarized in Table 1. The majority of the participants were female (71.1%), held a bachelor’s degree (59.4%), and were single (57.2%). The average age of participants was 30.75 ± 11.54 years, and the mean BMI was 25.47 ± 5.21. Approximately half of the participants were employed (50.6%), reported a monthly household income exceeding AED 20,000 (AED 1 = USD 0.27) (49.6%), and were classified as overweight or obese (48.7%). Furthermore, the population exhibited diverse regional representation, with approximately 38% from the Gulf Cooperation Council (GCC) countries, 24.1% from other Arab countries, 20.9% from Asian countries, and 17% distributed across Africa, Europe, and America.

### 3.2. Meat Consumption Pattern Among Consumers

The data presented in Table 2 outline the meat consumption habits of the study participants. It was found that 86% of the participants consumed all types of meat, including red meat, poultry, and seafood. Approximately 10% consumed only white meat (poultry). A smaller percentage, about 4%, adhered to pescatarian (1.6%), vegetarian (1.7%), or vegan (0.9%) habits. In terms of meat consumption during meals, nearly all respondents had meat dishes for lunch (90%) or dinner (60%). About 13% of the participants divided their meat consumption between breakfast (7%) and snacks (6%). A smaller proportion, 3%, stated that they do not consume meat in any meal.

### 3.3. Degree of Meat Attachment Among Consumers

Participant attitudes towards MA are outlined in Table 3, where their level of agreement with 14 statements related to their MA (positive bond with meat) is provided. The table displayed responses ranging from “Strongly disagree” to “Strongly agree” across different categories. Overall, this study found a strong positive attachment to meat among the study population. For instance, two-thirds of the participants expressed a strong affinity for meat (65%), loving meals with meat (67%), agreeing that “eating meat is one of the good pleasures in life” (66%), and that “a good steak is without comparison” (66%). Similarly, over half of the population (57%) believed that meat is irreplaceable in their diets, that they would feel sad if forced to stop eating meat (55%), and they do not picture themselves without eating meat regularly (55%). At least 3/4 of the participants believed that eating meat was “a natural and indisputable practice” (78%). Despite this predominant positive sentiment, an equally sizable proportion of respondents (76%) also expressed negative associations with meat, such as that eating meat reminds them of the death and suffering of animals, and that they felt bad when thinking of eating meat. For roughly 1/2 of respondents, eating meat reminded them of disease (52.3%). Moreover, over half (54%) of the participants agreed that they would feel fine with a meatless diet, while 48% of them disagreed with the statement that they would feel weak if they could not eat meat. The statement “To eat meat is disrespectful towards life and the environment” did not receive unanimous agreement, with a quarter of the participants remaining neutral and 31% somewhat agreeing with it. The items indicated with (R) in Table 3 were inversely re-coded. The total score was calculated to determine the participants’ global MAQ score, which had a mean value of 50.59 ± 10.24 out of 70 for our study population.

### 3.4. Categories of Meat Attachment Question (MAQ)

Participants were then categorized as having low, medium, or high attachment to meat based on IQR of the mean score of MAQ. Approximately half of the participants had a medium level of attachment to meat (49.7%), while a quarter were highly attached to meat (24.1%) and a quarter (26.2%) were considered as having low attachment to meat (Figure 1).

### 3.5. Consumer Willingness Towards Cultured Meat Consumption

Regarding the familiarity of respondents regarding cultured meat, it is notable that 59.3% of the participants lacked familiarity with the term “cultured meat”. Results suggest that people are somewhat willing to consume cultured meat, with the average score being 11.07 ± 4.76 out of a total possible score of 25 (Table 4). Our study revealed that while more than a third of the participants expressed a willingness to try cultured meat (36%), the percentage of those open to consuming it regularly decreased by 8%, dropping to 28% of the population. Approximately half of the respondents were unwilling to replace conventional meat with cultured meat (69%) or replace meat substitutes such as soy with cultured meat (49%). A substantial 63% of the participants expressed reluctance to pay more for cultured meat compared to conventional meat. Subsequently, the mean score of 11.07 ± 4.76 was then binary categorized into less willingness (53.7%) and more willingness (46.3%) towards cultured meat consumption.

### 3.6. Types of Products (Protein Sources) That Would Be Appealing to Consumers If They Were Produced Using Cultured Methods

Figure 2 provides a visual depiction of the protein dietary sources that participants expressed willingness to consume if produced through cultured methods. Nearly half of the respondents (47%) stated their refusal to consume any of the mentioned protein sources should they be cultured. Conversely, close to a third (29.6%) of the participants indicated a preference for cultured poultry as their protein source of choice.

### 3.7. Comparison of MAQ Score Among Sociodemographic Characteristics

The MAQ global scores, which indicate the level of MA, varied significantly across different sociodemographic groups, as illustrated in Table 5. The table is segmented into two parts for comparison based on the MAQ global score and the MAQ categories outlined in Table 3. The current study findings reveal notable statistical variances in MA among various nationalities and gender groups. Specifically, males exhibited a significantly higher level of MA compared to females (mean MAQ score = 52.31 ± 10.21, *p* = 0.01). A larger proportion of women demonstrated weaker positive associations with meat compared to their male counterparts, with approximately 74.2% of women displaying low MAQ scores, as opposed to only 25.8% of men (*p* < 0.001). Moreover, greater MA was notably observed among participants from Asian countries (mean MAQ score = 51.47 ± 9.54, *p* = 0.003), with a gradual decline in MAQ scores among respondents from other Arab countries, GCC, and African countries. Additionally, there was a significant increase in positive MA with higher BMI, as evidenced by the highest MAQ mean scores observed among obese participants (52.37 ± 9.17, *p* = 0.029). Furthermore, participants with varying educational backgrounds were categorized into significantly different MAQ groups, where approximately 51.1% of highly meat-attached participants held a bachelor’s degree, and 30% possessed a master’s/PhD degree (*p* = 0.007).

### 3.8. Comparison of Willingness Score Toward Cultured Meat Consumption Among Sociodemographic Characteristics

Table 6 presents participant scores of willingness to consume cultured meat segmented into categories. The data portray differences across various sociodemographic characteristics. No significant differences were observed in willingness scores across sociodemographic characteristics such as gender, employment status, marital status, household income, and BMI classification. However, younger people (<30 years old) had greater willingness scores compared to their older counterparts (11.50 ± 4.89 vs. 10.41 ± 4.48, *p* = 0.044). Participants with less formal education (diploma and below) tended to have higher willingness scores than university graduates (12.07 ± 5.00 vs. 10.84 ± 4.67; *p* = 0.016). Also, scores differed significantly between different nationalities, more specifically, participants of Arab nationality exhibited significantly higher willingness scores in comparison to participants of other non-Arab nationalities (11.54 ± 4.48 vs. 9.95 ± 5.20, *p* < 0.001).

### 3.9. Determinants of the Total MAQ and Willingness Towards Cultured Meat in the Study Population Depicted by Simple Linear Regression

Table 7 illustrates the crude model of the determinants of the MAQ and willingness toward cultured meat consumption total scores using simple linear regression. The analysis revealed that gender significantly predicted the MAQ total score, with males demonstrating an 11% higher likelihood of attaining higher MAQ scores in contrast to females (β = 0.107, 95% CI: −4.26, −0.58; *p* = 0.010). Sociodemographic traits, including age, educational level, and nationality, emerged as significant predictors reflecting the total score of willingness toward cultured meat consumption. Younger people were 11% more willing to consume cultured meat (β = 0.113, 95% CI: (−1.885, −0.303; *p* = 0.007), while university graduates demonstrated a 10% lower willingness than those from lower educational levels (β = 0.100, 95% CI: (−2.219, −0.227; *p* = 0.016). Arabs displayed a 15% higher likelihood of achieving higher willingness scores than non-Arabs (β = 0.152, 95% CI: (−2.428, −0.744; *p* < 0.001).

## 4. Discussion

This study was conducted to evaluate the extent of attachment to meat among consumers in the UAE and subsequently explore their willingness towards cultured meat consumption. The existing body of consumer research has been skewed towards Western markets [25], warranting an investigation into possible emerging markets in regions such as the UAE. Our findings revealed that a significant majority of respondents regularly consumed various types of conventional meat. Additionally, the study found that although participants had a strong attachment to meat, most were reluctant to embrace cultured meat. Males showed a stronger affinity to conventional meat than females. Participants more willing to consume cultured meat were younger, of lower educational backgrounds, and of Arab nationality. The demographic makeup of our study population included females (71%) and males (29%), with the highest representation observed in the 18–30-year-old participants. This observation is consistent with previous surveys in the UAE and other locations, where female respondents dominated the samples [46,47,48].

The current study’s findings revealed that the majority of consumers in the UAE consumed all types of meat, including red meat, poultry, and chicken (86%), while only 2.6% adhered to a strictly vegetarian/vegan diet. Research on the prevalence of veganism and vegetarianism varies by region, with estimates indicating that 16% of people in the African and Middle Eastern region (including Egypt, Morocco, Saudi Arabia, South Africa, and the United Arab Emirates) identify as vegetarians, while 6% identify as vegans [49]. A cross-sectional study conducted in Saudi Arabia found that the prevalence of veganism among Saudi nationals was 4.7%, while the rate of vegetarianism was 7.8% [50].

Research indicates that while a shift towards a more vegetarian-based diet leads to a host of benefits, high meat consumption, limited interest in meat substitutes, and hesitancy towards plant-based diets continue to reflect prevalent cultural norms in most societies worldwide [51]. Additionally, the UAE Ministry of Foreign Trade reported that the per-capita meat consumption in the UAE stood at 85.14 kg, which is 18 times higher than the global average. This elevated level of consumption can be attributed to a higher per-capita income (USD 55,000) and its influence on daily dietary practices [52].

When examining meat consumption patterns, around 86% of the participants consumed all kinds of meat, while 10% consumed only poultry meat. This observation is consistent with a study in the UAE that analysed meat demand, whereby poultry meat contributed nearly half (49%) of per-capita meat consumption, followed by red meats—lamb, goat, and beef (26%), as well as other meat varieties [52]. The majority of respondents tended to include meat dishes in their lunch meals (90%) followed by dinner (60%). This inclination can be attributed to the Middle Eastern tradition of family lunches as the primary meal of the day, a practice distinct from American cultures where dinner is considered the main meal. Research in the UAE indicates that a typical lunch often features traditional meat or vegetable stews, with rice as a staple accompaniment. It is common for many to forgo breakfast entirely, with some opting for a light dinner, thereby reserving meat-centric main courses for lunch [53].

The current study revealed an average MAQ score of 50.59 ± 10.24 (score percentage 72.27%) among participants, indicative of a significant attachment to meat. At least 2/3 of respondents identified themselves as meat enthusiasts (65%), expressed enjoyment of meat-based meals (67%), acknowledged meat consumption “as a source of pleasure” (66%), and considered a good steak as “unparalleled” (66%). These findings resonate with a comparable study in the UK, where 60.4% of participants exhibited high MA scores [54]. The survey revealed mixed feelings about meat consumption: 76% associated it with animal suffering, and 52.3% linked it to disease. Additionally, 54% felt they could be fine with a meatless diet, while 48% disagreed they would feel weak without meat. These findings underscore the coexistence of positive and negative associations with meat consumption. The discourse around meat consumption has increasingly focused on environmental sustainability, health concerns, and animal rights [55]. For instance, animal-based products exhibit higher environmental impacts compared to nutritionally equivalent plant-based foods [56]. However, the large majority of consumers exhibit an apparent disconnect between a desire to prevent animal suffering (indicating lower meat attachment) and the consumption of meat products, a phenomenon that has been termed the “meat paradox” [41]. This concept warrants consideration when examining meat attachment, its constructs, and its implications in determining perceptions toward cultured meat and alternative products.

The present study sought to evaluate consumer familiarity and acceptance of cultured meat. Almost 59% of participants were unfamiliar with the term “cultured meat” and moderately willing to consume cultured meat. Study findings showed that while 35% of participants expressed some willingness to try cultured meat, only 28% were clearly willing to purchase cultured meat regularly. These figures indicate a lower receptiveness than in similar studies from other countries. For instance, a study in Brazil evaluated public attitudes towards cultured meat amongst 408 consumers after a video was shown explaining the process of culturing meat from a feather originating from a broiler chicken. They found that despite the lower initial awareness of cultured meat, more than half (60.2%) of the respondents expressed the intent to consume cultured meat [47]. It is probable that the video shown to the Brazilian cohort in the previous study may have elicited emotions that influenced the participants’ inclination toward cultured meat. Consequently, the lower willingness observed in the current study might have resulted from the absence of additional visual information. In the present work, great care was taken to present the participants with unbiased and clear information regarding cultured meat to avoid priming and biasing responses. It is likely that the overall reluctance towards accepting cultured meat stems from the religious beliefs of study participants, as religious beliefs exert a significant influence on dietary decisions. For example, specific tenets of Islam impose restrictions on meat consumption. According to Islamic doctrine, halal food that encompasses items sanctioned under Islamic *Shariah* law is permissible, while haram food is prohibited [33]. Since the majority of the population in the UAE identifies as Muslim (76.9%) [43], it is likely that the study participants harboured concerns about the halal status of cultured meat. Some participants expressed unwillingness to consume cultured meat unless its halal compliance is validated. This observation aligns with the findings of a study in Singapore, where Muslim participants indicated that they would only consider consuming cultured meat when the provided regulatory authorities validate its halal status [57]. Members of the Muslim community rely on food regulatory authorities to assure them that cultured meat complies with the Islamic *Shariah* laws and is safe for consumption. Islamic scholars have contended that cultured meat can be deemed halal if the stem cells are procured from animals slaughtered in accordance with Islamic stipulations and if no blood or serum is employed in its manufacture [58]. The results of this study highlight a significant hesitancy towards adopting cultured meat alternatives. Specifically, 64% of participants expressed that they were unwilling to try cultured meat, and at least half of them (49%) preferred soy and other plant-based alternatives over cultured meat. An even higher percentage—72%—indicated that they would be reluctant to purchase cultured meat regularly or pay a premium for it. This finding contradicts previous studies where participants are willing to pay extra premiums for cultured meat. For instance, Rolland et al. [59] found that 58% of the European participants were willing to pay 37% above the price of conventional meat for cultured meat. Conversely, in the USA, while a majority of respondents expressed a willingness to sample cultured meat, only 30% indicated a definite or probable inclination to consume it on a regular basis and were less likely to pay a premium price for cultured meat [20].

According to the literature, a positive perspective toward cultured meat is influenced by three key factors. Firstly, consumers exhibited a favorable disposition when they were familiar with or received knowledge regarding the technology underpinning cultured meat [22,60]. Secondly, their attitude was positive following their exposure to pro-cultured meat information, encompassing aspects such as animal welfare and environmental benefits [47,61] or food safety and antibiotic resistance [59,62]. Thirdly, consumers with ethical considerations towards animals were more inclined to consider trying cultured meat [24,63]. However, certain studies presented conflicting evidence, suggesting that the environmental advantages that accompany cultured meat consumption hold lower significance in the food-related decision-making process for many consumers. Consequently, augmenting consumers’ awareness of these benefits may not necessarily lead to an increase in their acceptance [64].

As was observed in the current study, religion plays a pertinent role in food choices and, thus, is a vital consideration when consumers decide to consume cultured meat. Studies amongst Muslims in countries such as Singapore [57] and Britain [65] point to the public concern among Muslim consumers to validate cultured meat’s halal status under the Islamic *Shariah* and guarantee safety for human consumption [33].

Nearly half of the study participants (47%) refused to consume any cultured protein products originating from poultry, beef, mutton, fish, eggs, or dairy (Figure 2). Among the individuals open to consuming these cultured products, a significant majority favoured cultured poultry over beef and fish. This observation aligns with a study by Bryant et al., which indicated that among respondents in support of cultured products, 42.8% were willing to buy chicken, 42.6% would buy fish, and 39% beef [66]. In the UAE, poultry meat represented almost half (49%) of per-capita meat consumption, and this was followed by other red meats (26%) [52], thus substantiating the current results. Further investigation is warranted to obtain a more comprehensive understanding of consumer acceptance of various sources of cultured meat alternatives.

The present findings also demonstrated a notable correlation between participants’ BMI and their positive attachment to meat (r = 0.122, *p* = 0.003). Specifically, we observed that individuals with a higher BMI exhibited a stronger attachment to meat. Prior research suggests that an individual’s attachment to meat plays a pivotal role in determining their meat consumption levels, with higher meat consumption typically associated with a strong attachment [27,28]. Furthermore, evidence indicates a strong correlation between the increased prevalence of obesity and the widespread availability of meat in many countries [9,67].

The present findings showed that sociodemographic traits such as age, education, and nationality were predictive of willingness to accept cultured meat. Younger participants and those from a lower educational background were more willing to try cultured meat (mean willingness score = 11.50 ± 4.89; *p* = 0.044 and 0.100; *p* = 0.016, respectively). This contrasts with a study by Wilks et al. that did not find any significant influence of participant age or educational status on their willingness to consume cultured meat [20]. The results of other studies show that while most are not in favor of cultured meat technology, younger individuals are more open to trying it [25,68]. For instance, Mancini and Antonioli et al. [22] found that young, educated consumers who wanted to reduce meat intake were more prone to consume cultured meat. Similarly, Dupont and Fiebelkorn et al. [63] found that German children and adolescents held favorable views towards cultured food, and this was attributed to lower levels of food-related disgust. However, cultural context played a significant role in determining the willingness of participants towards cultured meat consumption. An interesting finding in the current study was that participants with less education seemed to be more willing to try cultured meat. This finding contrasts with most studies that found willingness to consume cultured meat increased with education level [24,59,61]. The more highly educated consumers in the current study may likely have had other reservations that could have reduced their willingness to accept cultured meat. For instance, a study involving educated students and scientists in France found that the vast majority of educated consumers viewed cultured meat negatively and wished to continue to eat conventional meat, with some consumers even opting to consume less meat in response to environmental concerns [69]. The current study also showed that respondents of Arabic ethnicity demonstrated a significantly higher willingness score compared to other populations. Most consumer studies on cultured meat are based on Western markets, while data on Arab consumers are lacking. Bryant et al. [40] conducted a cross-country analysis of consumer perceptions of cultured meat alternatives in the USA, India, and China. Through online surveys, they calculated the mean likelihood score towards purchasing cultured meat. The mean likelihood for Indian consumers was 3.52 ± 1.30, and for Chinese consumers, it was 3.52 ± 1.14 out of a total score of 5. In comparison, consumers from the USA had a lower mean score of 2.72 ± 1.35. Siegrist and Hartmann [70] conducted a web-based study across 10 countries (USA, Sweden, Australia, China, Germany, Mexico, France, South Africa, and England) among 6128 participants. The findings revealed significant cross-cultural differences in the acceptance of cultured meat. Specifically, it was noted that French consumers exhibited substantially lower acceptance compared to consumers in other countries [70]. Hence, it is crucial to consider the cultural context when analysing the variables influencing consumer acceptance of cultured meat.

The results of the present study reveal that gender played a significant role in influencing an individual’s meat attachment. Male consumers in the UAE showed an 11% higher likelihood of achieving higher MAQ scores compared to females (OR = 0.107, 95% CI: −4.26, −0.58; *p* = 0.010). This finding is corroborated with results from other studies, including one conducted in Germany that explored gender-based disparities in meat consumption. The study found that the average German woman consumed less red meat than her male counterpart. Furthermore, men exhibited a stronger meat attachment and were less inclined to lower meat consumption compared to women. Findings revealed that sex was a significant predictor, i.e., females showed higher levels of intention (β = 0.17, *p* < 0.001) and willingness (β = 0.18, *p* < 0.001) to reduce meat consumption than males [71]. This result suggests that communication strategies aimed at influencing meat consumption should prioritize the modification of meat attachment and tailor messages to the different genders [72]. Additionally, prior research corroborates this finding, indicating that females unfamiliar with cultured meat and vegan consumers exhibited the most significant positive shift in perceptions following exposure to information about the safety aspects of cultured meat [60]. Concurrently, Tucker et al. observed that women were more inclined to harbor an overall negative attitude toward cultured meat, with 69% of individuals expressing a positive view being male [73]. Understanding consumer attitudes toward cultured meat necessitates considering various factors, including sociodemographic characteristics, as well as the underlying cultural context.

The findings of this study reveal several key barriers faced by the target population. Firstly, 59% of respondents reported being unfamiliar with the term “cultured meat”, which highlights a concerning lack of awareness and trust surrounding these alternatives, as well as apprehensions regarding their safety for consumption. This lack of awareness likely contributes to the hesitance, with 47% of participants stating that they would refuse to consume any cultured alternative when it becomes available. Secondly, cost is identified as a significant obstacle, as the majority of participants expressed unwillingness to pay an additional premium for cultured meat. Lastly, it is important to recognize that cultural and religious sentiments among participants, particularly within the Muslim community, appear to influence their consumer choices, as supported by findings from several studies [57,65].

To address the inherent challenges associated with perceptions of cultured alternatives, it is essential to propose specific strategies for key stakeholders, including companies manufacturing cultured products, government entities, and relevant associations. First, there is a pressing need to increase awareness, understanding and knowledge of cultured alternatives, their production processes, and their proposed benefits to improve overall public perceptions through targeted educational campaigns for cultured meat technology. Future research efforts that concentrate on the composition of cultured meat products are warranted. Additionally, it is recommended that studies adopt a data-driven approach to evaluate and demonstrate the safety of cultured meat. Such efforts can significantly support consumer acceptance and facilitate the full realization of these products’ potential. Public acceptance is fundamentally contingent upon the assumption that cultured products are considered safe for consumption by food safety regulatory authorities [15,16,25]. Hence, rigorous safety assessments are imperative for cultured products, which will help establish clear standards to safeguard public health and consumer interests within the Arab region. The study results could thus guide both public policies, as well as the regulation of activities related to cultured meat in the coming years, professional orders, private businesses, and the general public. Regulatory agencies elsewhere have developed broad guidance for safety assessments, such as the Singapore Food Agency, which has granted approval for the sale of cultured chicken products in Singapore [17]. The United Arab Emirates is actively exploring viable cultured alternatives, facilitated by collaborations between AGWA (AgriFood Growth & Water Abundance) and Believer Meats, an American company operating within the cultivated meat industry. AGWA will also collaborate with Believer Meats to establish a regulatory framework and halal certification standards for cultured meat products, which will contribute to the development of a reliable model for the entry and expansion of cultured meat in the regional meat markets [74]. Second, effective marketing strategies must be implemented to enhance consumer acceptance and outreach. It is noteworthy that consumers’ willingness to purchase cultured meat is predominantly influenced by personal factors such as affordability and taste over societal considerations [25]. This insight is particularly relevant for industry stakeholders and policymakers as they formulate optimal pricing and marketing strategies for cultured alternatives. Ensuring that cultured meat products are available at affordable prices for the general public will serve as a significant incentive for acceptance. Additionally, ensuring that cultured meat closely resembles conventional meat in terms of organoleptic characteristics will further foster positive perceptions.

Finally, obtaining halal certifications from recognized organizations is essential to clarify any uncertainties regarding the consumption of cultured meat in accordance with Islamic *Shariah* amongst a Muslim-majority population [33,65]. There is a critical need for legislative measures and religious oversight to enhance the trust and acceptance of cultured meat alternatives before commercialization.

The present study stands out as the first to delve into consumer perceptions of novel food technology, which is the culturing of meat, within the UAE. Previous research has overlooked emerging markets such as the UAE, disproportionately focusing primarily on the West [25]. However, it is essential to acknowledge the limitations inherent in our study, using a self-reported questionnaire, the snowball sampling method, and the cross-sectional study design. Another potential limitation was the overrepresentation of females among the respondents, a recurring trend in online questionnaires [75]. Nevertheless, all analyses in this paper were conducted after controlling for confounding factors, ensuring the validity of the results presented. Utilizing an online survey enabled the collection of data from all seven emirates in the UAE and ensured participant anonymity, thereby reducing social desirability bias.

## 5. Conclusions

This study explores meat attachment and the participants’ willingness to consume cultured meat by consumers in the UAE. While most participants exhibited a substantial level of meat attachment or appreciation, there was also reluctance towards embracing cultured meat. Discernible variations in meat attachment levels and willingness to consider cultured meat were identified across different nationalities and genders. Participants who displayed a higher inclination towards cultured meat consumption were younger, less educated, and of Arabic ethnicity. Males exhibited a stronger attachment to meat compared to females. Overall, the current study provides insight that can serve as a guide for regulatory and industry stakeholders to enhance public acceptance and support successful commercialization. It is beneficial to proactively address ethical, environmental, human health, and cultural considerations. Furthermore, engaging with consumer perceptions related to taste and affordability can provide valuable insights that contribute to overall success in the market. Leveraging the trust that consumers place in government institutions for determining the safety and halal status of cultured meat is important since these institutions can play a pivotal role in disseminating accurate scientific information on the nutritional and safety characteristics of cultured meat.

## Figures and Tables

**Figure 1 nutrients-17-00028-f001:**
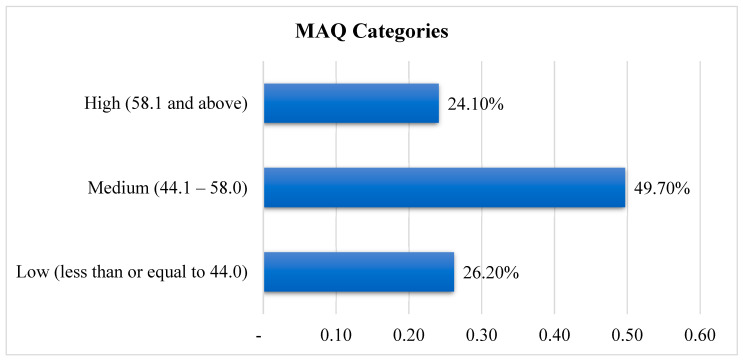
Categories of MAQ.

**Figure 2 nutrients-17-00028-f002:**
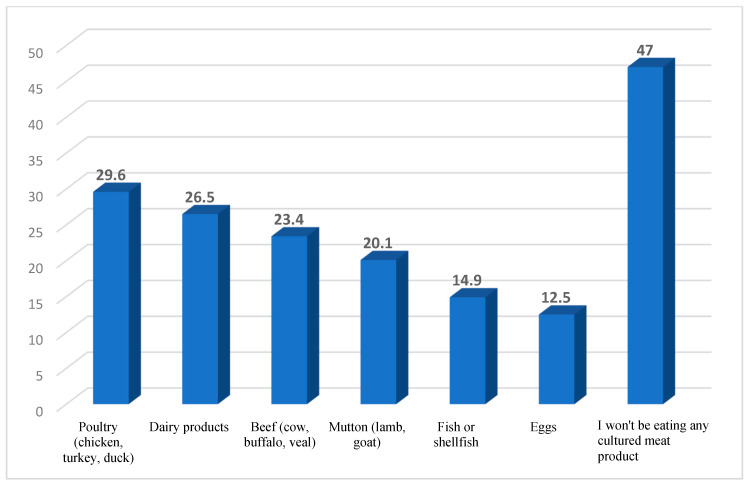
Types of products (protein sources) that would be appealing to consumers if they were produced using cultured methods.

**Table 1 nutrients-17-00028-t001:** Sociodemographic characteristics of the study population (*n* = 577).

Sociodemographic Variable	Categories	*n*	%
Sex	Male	167	28.9
	Female	410	71.1
Age	30.75 ± 11.54 years *		
Age categories	<30 years old	349	60.5
	30–50 years old	186	32.2
	>50 years old	42	7.3
Educational level	Diploma and below	107	18.5
	BSc	343	59.4
	MSc/PhD	127	22.0
Employment status	Employed	292	50.6
	Unemployed	89	15.4
	Student	196	34.0
Marital status	Single	330	57.2
	Ever married	247	42.8
Nationality	Gulf Cooperation Council (GCC)	219	38.0
	Other Arabs	139	24.1
	Asia	121	20.9
	Africa, Europe, and America	98	17.0
Monthly household income (AED) **	<10,000	149	25.8
	10,000–20,000	142	24.6
	20,000–30,000	139	24.1
	>30,000	147	25.5
BMI	25.47 ± 5.21 *		
BMI category ^a^ (kg/m^2^)	Underweight (<18.0)	36	6.2
	Normal (18.0–24.9)	260	45.1
	Overweight (25.0–29.9)	178	30.8
	Obese (>30)	103	17.9

* Mean ± standard deviation. ** 1 AED = 0.27 USD. ^a^ BMI was categorized as less than 18.5 kg/m^2^ (underweight), 18.5 kg/m^2^ to less than 25.0 kg/m^2^ (normal weight), 25.0 kg/m^2^ to less than 30.0 kg/m^2^ (overweight), and 30.0 kg/m^2^ or more (obesity) [45].

**Table 2 nutrients-17-00028-t002:** Meat consumption pattern among consumers in the UAE (*n* = 577).

		*n*	%
Meat consumption pattern	Meat-eating (red meat, poultry, seafood)	497	86.1
	Eat white meat only (poultry—chicken, turkey)	56	9.7
	Pescatarian (only seafood)	9	1.6
	Vegetarian and vegan	15	2.6
Meat consumption in meals (multiple choice question)	Breakfast	39	6.8
	Lunch	520	90.1
	Dinner	347	60.1
	Snacks	32	5.5
	I do not eat meat in any meal	16	2.8

**Table 3 nutrients-17-00028-t003:** Degree of meat attachment among consumers (*n* = 577).

MAQ Items Included	MAQ Statements	Strongly Disagree *n* (%)	Somewhat Disagree *n* (%)	Neutral *n* (%)	Somewhat Agree *n* (%)	Strongly Agree *n* (%)
MAQ-1	To eat meat is one of the good pleasures in life	29 (5.1)	40 (6.9)	123 (21.3)	261 (45.2)	124 (21.5)
MAQ-2	Meat is irreplaceable in my diet	38 (6.6)	99 (17.2)	112 (19.4)	208 (36.0)	120 (20.8)
MAQ-3	I feel bad when I think of eating meat (R)	18 (3.1)	39 (6.8)	80 (13.9)	184 (31.9)	256 (44.4)
MAQ-4	I love meals with meat	25 (4.3)	35 (6.1)	128 (22.2)	252 (43.7)	137 (23.7)
MAQ-5	To eat meat is disrespectful towards life and the environment (R)	39 (6.8)	76 (13.2)	142 (24.6)	179 (31.0)	141 (24.4)
MAQ-6	I’m a big fan of meat	23 (4.0)	112 (19.4)	65 (11.3)	244 (42.3)	133 (23.1)
MAQ-7	If I couldn’t eat meat, I would feel weak	78 (13.5)	198 (34.3)	140 (24.3)	123 (21.3)	38 (6.6)
MAQ-8	If I were forced to stop eating meat, I would feel sad	43 (7.5)	101 (17.5)	115 (19.9)	201 (34.8)	117 (20.3)
MAQ-9	Meat reminds me of diseases (R)	28 (4.9)	91 (15.8)	156 (27.0)	198 (34.3)	104 (18.0)
MAQ-10	By eating meat, I’m reminded of the death and suffering of animals (R)	11 (1.9)	44 (7.6)	85 (14.7)	169 (29.3)	268 (46.4)
MAQ-11	Eating meat is a natural and undisputable practice	18 (3.1)	21 (3.6)	83 (14.4)	258 (44.7)	197 (34.1)
MAQ-12	I don’t picture myself without eating meat regularly	40 (6.9)	97 (16.8)	122 (21.1)	178 (30.8)	140 (24.3)
MAQ-13	I would feel fine with a meatless diet (R)	36 (6.2)	95 (16.5)	132 (22.9)	180 (31.2)	134 (23.2)
MAQ-14	A good steak is without comparison	29 (5.0)	46 (8.0)	124 (21.5)	237 (41.1)	141 (24.4)
MAQ total score = 50.59 ± 10.24						

The items marked with (R) in the table were inversely coded.

**Table 4 nutrients-17-00028-t004:** Consumer willingness towards cultured meat consumption (*n* = 577).

		Not Willing at All *n* (%)	Somewhat Willing *n* (%)	Moderately Willing *n* (%)	Very Willing *n* (%)	Absolutely Willing *n* (%)
Willingness to try cultured meat		128 (22.2)	114 (19.8)	131 (22.7)	125 (21.7)	79 (13.6)
Willingness to purchase cultured meat regularly		192 (33.3)	105 (18.2)	117 (20.3)	99 (17.2)	64 (11.0)
Willingness to eat cultured meat as a replacement for conventional meat		275 (47.7)	125 (21.7)	112 (19.4)	37 (6.4)	28 (4.8)
Willingness to pay a higher price for cultured meat than conventional meat		362 (62.7)	93 (16.1)	75 (13.0)	36 (6.2)	11 (2.0)
Willingness to eat cultured meat as a replacement for meat substitutes (i.e., made from soy)		281 (48.7)	118 (20.5)	102 (17.7)	50 (8.7)	26 (4.4)
Willingness mean score		11.07 ± 4.76 *				
Willingness categories (mean score)	Less willingness (<11.07)	310 (53.7%)				
	More willingness (>11.07)	267 (46.3%)				

* Mean ± standard deviation.

**Table 5 nutrients-17-00028-t005:** Comparison of MAQ score among sociodemographic characteristics (*n* = 577).

Sociodemographic Variable		*p*-Value Comparison by Total Score Mean ± SD	Low MAQ *n* (%)	Medium MAQ *n* (%)	High MAQ *n* (%)	*p*-Value Comparison by Categories
Sex	Male (*n* = 167)	52.31 ± 10.21	39 (25.8)	68 (23.7)	60 (43.2)	**<0.001**
	Female (*n* = 410)	49.89 ± 10.17	112 (74.2)	219 (76.3)	79 (56.8)	
		***p* = 0.010**				
Age	<30 years old	50.81 ± 10.15	86 (57.0)	182 (63.4)	81 (58.3)	0.191
	30–50 years old	50.81 ± 10.13	48 (31.8)	88 (30.7)	50 (36.0)	
	>50 years old	47.79 ± 11.18	17 (11.3)	17 (5.9)	8 (5.8)	
		*p* = 0.184				
Educational level	Diploma and below	51.18 ± 9.49	27 (17.9)	56 (19.5)	24 (17.3)	**0.007**
	BSc	50.13 ± 10.07	87 (57.6)	185 (64.5)	71 (51.1)	
	MSc/PhD	51.32 ± 11.25	37 (24.5)	46 (16.0)	44 (31.7)	
		*p* = 0.428				
Employment status	Employed (*n* = 292)	50.58 ± 10.86	80 (53.0)	136 (47.4)	76 (54.7)	0.360
	Unemployed (*n* = 89)	50.00 ± 9.87	26 (17.2)	42 (14.6)	21 (15.1)	
	Students (*n* = 196)	50.86 ± 9.44	45 (29.8)	109 (38.0)	42 (30.2)	
		*p* = 0.805				
Marital status	Single (*n* = 330)	50.86 ± 9.95	83 (55.0)	171 (59.6)	76 (54.7)	0.513
	Ever married (*n* = 247)	50.22 ± 10.61	68 (45.0)	116 (40.4)	63 (45.3)	
		*p* = 0.456				
Nationality	GCC (*n* = 219)	50.68 ± 9.71 ^a^	59 (39.1)	111 (38.7)	49 (35.3)	0.094
	Levant (*n* = 139)	51.35 ± 9.95 ^a^	35 (23.2)	66 (23.0)	38 (27.3)	
	Asia (*n* = 121)	51.47 ± 9.54 ^a^	22 (14.6)	69 (24.0)	30 (21.6)	
	Africa, Europe, and America (*n* = 98)	47.56 ± 12.65 ^b^	35 (23.1)	41 (14.3)	22 (15.9)	
		***p* = 0.003**				
Monthly household income (AED)	<10,000 (*n* = 149)	50.98 ± 10.39	49 (32.4)	65 (22.7)	35 (25.2)	0.377
	10,000–20,000 (*n* = 142)	51.45 ± 9.46	29 (19.2)	81 (28.2)	32 (23.0)	
	20,000–30,000 (*n* = 139)	50.94 ± 9.62	34 (22.5)	72 (25.1)	33 (23.7)	
	>30,000 (*n* = 147)	50.31 ± 11.39	39 (25.8)	69 (24.0)	39 (28.1)	
		*p* = 0.701				
BMI categories	Underweight (*n* = 36)	46.58 ± 12.52 ^a^	14 (9.3)	19 (6.6)	3 (2.2)	0.078
	Normal (*n* = 260)	50.26 ± 10.23 ^b^	72 (47.7)	127 (44.3)	61 (43.9)	
	Overweight (*n* = 178)	50.84 ± 10.15 ^b^	47 (31.1)	83 (28.9)	48 (34.5)	
	Obese (*n* = 103)	52.37 ± 9.17 ^b^	18 (11.9)	58 (20.2)	27 (19.4)	
		***p* = 0.029 ***				

* Continuous BMI value was significantly correlated with the MAQ total score (r = 0.122, *p* = 0.003). ^a, b^ = for the significant variables with more than two categories, the different letters indicate significant difference. The numbers in bold are statistically significant. For all data, the significance level was set at *p* < 0.05.

**Table 6 nutrients-17-00028-t006:** Comparison of willingness score toward cultured meat consumption among sociodemographic characteristics (*n* = 577).

Sociodemographic Variable		*p*-Value Comparison by Total Score Mean ± SD	Less Willingness *n* (%)	More Willingness *n* (%)	*p*-Value Comparison by Categories
Sex	Male (*n* = 167)	11.15 ± 5.17	89 (28.7)	78 (29.2)	0.483
	Female (*n* = 410)	11.04 ± 4.58	221 (71.3)	247 (70.8)	
		*p* = 0.796			
Age	≤30 years old (*n* = 349)	11.50 ± 4.89	177 (57.1)	172 (64.4)	**0.044**
	>30 years old (*n* = 229)	10.41 ± 4.48	133 (42.9)	95 (35.6)	
		***p* = 0.005**			
Educational level	Diploma and below (*n* = 107)	12.07 ± 5.00	47 (15.2)	60 (22.5)	**0.016**
	University graduate (*n* = 470)	10.84 ± 4.67	263 (84.8)	207 (77.5)	
		***p* = 0.022**			
Employment status	Employed (*n* = 292)	11.01 ± 4.64	158 (51.0)	134 (50.2)	0.752
	Unemployed (*n* = 89)	10.91 ± 5.10	49 (15.8)	40 (15.0)	
	Students (*n* = 196)	11.22 ± 4.78	103 (33.2)	93 (34.8)	
		*p* = 0.841			
Marital status	Single (*n* = 330)	11.40 ± 4.91	171 (55.2)	159 (59.6)	0.410
	Ever married (*n* = 247)	10.62 ± 4.51	139 (44.8)	108 (40.4)	
		*p* = 0.051			
Nationality	Arab (*n* = 406)	11.54 ± 4.48	200 (64.5)	206 (77.2)	**<0.001**
	Non-Arab (*n* = 171)	9.95 ± 5.20	110 (35.5)	61 (22.8)	
		***p* = < 0.001**			
Monthly household income (AED)	<10,000 (*n* = 149)	11.09 ± 4.59	74 (23.9)	75 (28.1)	0.506
	10,000–20,000 (*n* = 142)	10.88 ± 4.81	78 (25.2)	64 (24.0)	
	20,000–30,000 (*n* = 139)	10.92 ± 4.89	81 (26.1)	58 (21.7)	
	>30,000 (*n* = 147)	11.37 ± 4.77	77 (24.8)	70 (26.2)	
		*p* = 0.810			
BMI categories	Underweight (*n* = 36)	11.47 ± 5.05	17 (5.5)	19 (7.1)	0.101
	Normal (*n* = 260)	10.66 ± 4.63	153 (49.4)	107 (40.1)	
	Overweight (*n* = 178)	10.99 ± 4.76	93 (30.0)	85 (31.8)	
	Obese (*n* = 103)	12.10 ± 4.85	47 (15.2)	56 (21.0)	
		*p* = 0.070			

The numbers in bold are statistically significant. For all data, the significance level was set at *p* < 0.05.

**Table 7 nutrients-17-00028-t007:** Determinants of the total MAQ and willingness towards cultured meat in the study population depicted by simple linear regression (*n* = 577).

Variable	MAQ Total Score	Willingness Total Score
**Sex (Ref: female)** Male	0.107 (−4.26, −0.58) ***p* = 0.010**	0.011 (−0.970, 0.745) *p* = 0.796
**Age (Ref: >30 years)** ≤30 years	0.027 (−2.27, 1.16) *p* = 0.525	0.113 (−1.885, −0.303) ***p* = 0.007**
**Educational level: (Ref: university)** Diploma and below	0.028 (−2.88, 1.43) *p* = 0.508	0.100 (−2.219, −0.227) ***p* = 0.016**
**Employment status (Ref: employed)** Unemployed	(−1.66, 1.69) *p* = 0.987	0.019 (−0.331, 0.529) *p* = 0.652
**Marital status (Ref: Single)** Ever married	0.031 (−2.334, 1.050) *p* = 0.456	0.081 (−1.563, 0.004) *p* = 0.051
**Nationality (Ref: Non-Arab)** Arab	0.029 (−2.492, 1.176) *p* = 0.481	0.152 (−2.428, −0.744) ***p* < 0.001**
**Household income (ref: less than 20,000 AED)** 20,000 AED and above	0.006 (−1.562, 1.788) *p* = 0.895	0.021 (−0.254, 0.435) *p* = 0.608
**BMI (Ref: Normal and underweight)** Overweight and obese	0.078 (−0.080, 3.262) *p* = 0.062	0.079 (−0.015, 0.899) *p* = 0.058

Abbreviations: BMI: Body mass index, The numbers in bold are statistically significant. For all data, the significance level was set at *p* < 0.05.

## Data Availability

The original contributions presented in this study are included in the article. Further inquiries can be directed to the corresponding author(s).

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
