# Peer review of "Attachment to Meat and Willingness Towards Cultured Alternatives Among Consumers: A Cross-Sectional Study in the UAE"

_nutrients, 2024, doi:10.3390/nu17010028_

Round 1

Reviewer 1 Report

Comments and Suggestions for Authors

The introduction it is well written and revels the current state of the research field. The purpose of the work and its significance, is well defined. Materials and Methods are described with sufficient detail. The Study questionnaire/survey comprise Sociodemographic questions meat consumption pattern, Meat Attachment Questionnaire, willingness toward cultured meat consumption. Results Provide a concise and precise description of the experimental results, their interpretation as well as the experimental conclusions that can be drawn. The discussion of the results and how they can be interpreted in perspective of previous studies are well made and supported by appropriated bibliography. 

Author Response

Comment 1: The introduction it is well written and revels the current state of the research field. The purpose of the work and its significance, is well defined. Materials and Methods are described with sufficient detail. The Study questionnaire/survey comprise Sociodemographic questions meat consumption pattern, Meat Attachment Questionnaire, willingness toward cultured meat consumption. Results Provide a concise and precise description of the experimental results, their interpretation as well as the experimental conclusions that can be drawn. The discussion of the results and how they can be interpreted in perspective of previous studies are well made and supported by appropriated bibliography. 

Response: Thank you for your extensive review and appreciated comments. We really appreciate your positive feedback.

Reviewer 2 Report

Comments and Suggestions for Authors

The manuscript offers an insightful examination of meat attachment and consumer willingness towards cultured meat in the UAE. The article is well-structured and provides a comprehensive overview of the topic, utilizing appropriate statistical methods and presenting results systematically. However, there are several areas where clarity, rigor, and depth could be enhanced.

he article's purpose is clear: to investigate consumer perceptions of cultured meat and their attachment to conventional meat in the UAE. However, it could benefit from sharper focus and practical relevance. While the objectives are outlined in the abstract (lines 21-42) and introduction (lines 46-130), the link between findings and actionable outcomes—such as policy recommendations or consumer education strategies—is weak.

The study population is predominantly female (71.1%, line 247). This skew raises concerns about gender representativity, particularly given that gender is a significant variable influencing meat attachment. The authors should address how this imbalance might affect the study's conclusions or consider conducting a follow-up study with a more balanced gender representation.

While the discussion acknowledges religious concerns (lines 462-475), it fails to delve deeply into the role of halal certification in shaping consumer perceptions. Given the Muslim-majority context, a more detailed exploration of religious and ethical considerations could provide richer insights.

 Please articulate how the results will be applied, such as in marketing or public policy.

Table 4: shows percentages for different levels of willingness (e.g., 36% willing to try cultured meat), the authors provide minimal interpretation of these findings. For instance, they do not explore why only 28% are willing to purchase it regularly, missing an opportunity to discuss specific barriers (e.g., cultural, economic, or educational).

Nearly half of respondents (47%) refused to consume any cultured protein products, but this critical finding is not sufficiently discussed. The authors fail to address whether this reluctance is due to religious beliefs, lack of awareness, or other factors.

The conclusion does not fully address the implications of significant results, such as the 59.3% unfamiliarity with cultured meat (line 305) or the role of religious concerns (lines 462-475). These are central to the study's objectives but are underemphasized. Also the conclusions does not provide specific recommendations, despite results indicating clear barriers like cost concerns (63% unwilling to pay more, line 313) and cultural hesitations.

Author Response

Comment 1: The manuscript offers an insightful examination of meat attachment and consumer willingness towards cultured meat in the UAE. The article is well-structured and provides a comprehensive overview of the topic, utilizing appropriate statistical methods and presenting results systematically. However, there are several areas where clarity, rigor, and depth could be enhanced. The article's purpose is clear: to investigate consumer perceptions of cultured meat and their attachment to conventional meat in the UAE. However, it could benefit from sharper focus and practical relevance. While the objectives are outlined in the abstract (lines 21-42) and introduction (lines 46-130), the link between findings and actionable outcomes—such as policy recommendations or consumer education strategies—is weak.

Response 1: Thank you for your invaluable comment. According to the reviewer comments, the links between study results and actionable outcomes primarily, consumer education, marketing strategies, and policy recommendations were added to discussion section (Lines 514 – 545).

Comment 2. The study population is predominantly female (71.1%, line 247). This skew raises concerns about gender representativity, particularly given that gender is a significant variable influencing meat attachment. The authors should address how this imbalance might affect the study's conclusions or consider conducting a follow-up study with a more balanced gender representation.

Response 2: We are thankful to the reviewer for pointing this out. We agree with the reviewers’ suggestion to conduct a future follow up study with a balanced gender distribution. With that, the higher proportion of female respondents in research as compared to males, reflected in our study, has also been frequently reported in literature. A study by Saleh et al. found that females were more likely to respond to online surveys, attributing this to factors such as interest in the survey topic and perceived relevance (Saleh, A., & Bista, K. (2017). Examining factors impacting online survey response rates in educational research: Perceptions of graduate students. Online Submission13(2), 63-74.). Similarly, Sax et al. reported that females demonstrated higher response rates in web-based surveys, suggesting that women may be more inclined to engage in research participation (Sax, Linda J., Shannon K. Gilmartin, and Alyssa N. Bryant. "Assessing response rates and nonresponse bias in web and paper surveys." Research in higher education 44 (2003): 409-432.). As per the reviewer comment we have added a statement as a limitation in the discussion section (Lines 551 – 554).

Comment 3. While the discussion acknowledges religious concerns (lines 462-475), it fails to delve deeply into the role of halal certification in shaping consumer perceptions. Given the Muslim-majority context, a more detailed exploration of religious and ethical considerations could provide richer insights.

Response 3: Thank you again. As per your comments, we have added details exploration of religious considerations to draw attention to the role of halal certification in shaping consumer perceptions in lines 395 – 408, Lines 429 – 433, Lines 507 – 509 and Lines 542 – 545.

Comment 4. Please articulate how the results will be applied, such as in marketing or public policy.

Response 4: We are thanking you again. As per the comments, we have articulate the application of our findings in marketing and public policy from Lines 520 – 541.

Comment 5. Table 4: shows percentages for different levels of willingness (e.g., 36% willing to try cultured meat), the authors provide minimal interpretation of these findings. For instance, they do not explore why only 28% are willing to purchase it regularly, missing an opportunity to discuss specific barriers (e.g., cultural, economic, or educational).

Response 5: Thank you for pointing this. Accordingly, we have discussed specific barriers (cultural, economic and educational) which influence the willingness to purchase cultured meat (Lines 501 – 513)

Comment 6: Nearly half of respondents (47%) refused to consume any cultured protein products, but this critical finding is not sufficiently discussed. The authors fail to address whether this reluctance is due to religious beliefs, lack of awareness, or other factors.

Response 6: We are thanking you for this comment. Accordingly, we have discussed specific barriers religious and lack of knowledge as indicated in the aforementioned lines that influence the consumption of cultured protein alternatives. Religious barriers have been discussed in (Lines 395 – 408) and lack of awareness (Lines 382 – 384, 504 - 505)

Comment 7: The conclusion does not fully address the implications of significant results, such as the 59.3% unfamiliarity with cultured meat (line 305) or the role of religious concerns (lines 462-475). These are central to the study's objectives but are underemphasized. Also the conclusions does not provide specific recommendations, despite results indicating clear barriers like cost concerns (63% unwilling to pay more, line 313) and cultural hesitations.

Response 7: We are grateful to your extensive feedback in refining the quality of our work. We have modified the conclusion section following the reviewer suggestion (Lines 564 – 571).

Reviewer 3 Report

Comments and Suggestions for Authors

General comments:

The manuscript presents a market survey conducted in the United Arab Emirates on current meat consumption and the willingness among the population to eat cultured meat. Although in general, data obtained through unsupervised online questionnaires are always less valid from a scientific point of view, the high number of responses (577) helps to reduce the unreliability of the data obtained.

In general, the manuscript shows results that are in line with available knowledge, such as men showing a greater predisposition to meat consumption than women, or younger people showing a greater predisposition to consume cultured meat than older people. Other data are relatively more novel, such as the fact that people with less education are more willing to eat cultured meat.

The presentation of the manuscript is very good, with an adequate number of tables and figures, and a friendly and easy-to-read writing style (more so in the first part than towards the end). Overall the work is interesting and I think it could be published, although there are some things that could be substantially improved.

Specific comments:

1) Much is said in the manuscript about the adverse environmental effects of livestock production. Often these data are biased from a political point of view and are not entirely true. For example, cattle are often credited with using land on which wild ruminants actually live, not taking into account the reduction of forest fires in areas used by livestock, or not taking into account the enormous number of by-products that originate in the production of plant food for humans, which are then consumed by livestock. While it is not my intention to polemicize with the authors about this, I would appreciate a more neutral discourse on this issue in the manuscript.

2) Other factors that condition the consumption of cultured meat are its poor organoleptic properties. There is no mention of this in the text, and I think it would be good to point this out.

Line 38: Significant levels in abstract are not necessary.

Lines 111-116: I understand that this work is done in a Muslim country, but since the discussion section makes mention of Kosher meat according to Judaism, I think it is consistent to mention it in the introduction as well (or else delete its reference in the discussion).

Line 198: “reFmoved”. Please correct it.

Line 253: “GCC” is afterwards defined in line 342. Please define it the first time that appears in the main text.

Tables. Usually, the number of samples or observations is cited as “n”, more than “N”. In Table 2 is cited as “n”, and in other Tables as “N”. Please, be consistent.

Lines 296-298: MAQ is defined in subheading. Usually, in subheadings it is not recommended to use abbreviations. However, if it should be defined the first time it appears in the main text (line 298)

Lines 421-423: This is not all true. It is true in North American Cultures, but in Europe in many countries the main meal is the midday meal and not the evening meal.

Lines 506-509: nearly….fish” This rate is not according with those stablished in table 2. Please revise it.

References: Some sources are not cited according to MDPI guidelines. In example, “Meat Science” should be cited as “Meat Sci.”

Author Response

Comment 1: The manuscript presents a market survey conducted in the United Arab Emirates on current meat consumption and the willingness among the population to eat cultured meat. Although in general, data obtained through unsupervised online questionnaires are always less valid from a scientific point of view, the high number of responses (577) helps to reduce the unreliability of the data obtained. In general, the manuscript shows results that are in line with available knowledge, such as men showing a greater predisposition to meat consumption than women, or younger people showing a greater predisposition to consume cultured meat than older people. Other data are relatively more novel, such as the fact that people with less education are more willing to eat cultured meat. The presentation of the manuscript is very good, with an adequate number of tables and figures, and a friendly and easy-to-read writing style (more so in the first part than towards the end). Overall the work is interesting and I think it could be published, although there are some things that could be substantially improved.

Response 1: We are thanking you for the appreciated feedback.

Specific comments: 

Comment 2: Much is said in the manuscript about the adverse environmental effects of livestock production. Often these data are biased from a political point of view and are not entirely true. For example, cattle are often credited with using land on which wild ruminants actually live, not taking into account the reduction of forest fires in areas used by livestock, or not taking into account the enormous number of by-products that originate in the production of plant food for humans, which are then consumed by livestock. While it is not my intention to polemicize with the authors about this, I would appreciate a more neutral discourse on this issue in the manuscript.

Response 2: We are thanking you for highlighting this point. In order to adjust for a more neutral outlook, we have made changes in the manuscript in Line 61-62, Line 63-65, Line 74 – 77 as per the reviewer comment.

Comment 3: Other factors that condition the consumption of cultured meat are its poor organoleptic properties. There is no mention of this in the text, and I think it would be good to point this out. 

Response 3: Thank you for your excellent suggestion. As indicated, utilizing the organoleptic properties as way to increase cultured meat acceptance has been added in Lines 79 – 82 and Lines 534 – 541.

Comment 4: Line 38: Significant levels in abstract are not necessary.

Response 4: Thank you. The text was modified as per the reviewer’s recommendations (Line 50).

Comment 5: Lines 111-116: I understand that this work is done in a Muslim country, but since the discussion section makes mention of Kosher meat according to Judaism, I think it is consistent to mention it in the introduction as well (or else delete its reference in the discussion).

Response 5: Agree and thank you. The text is modified.

Comment 6: Line 198: “reFmoved”. Please correct it.

Response 6: Thank you again. The word is corrected.

Comment 7: Line 253: “GCC” is afterwards defined in line 342. Please define it the first time that appears in the main text.

Response 7: Thank you for pointing this. The text was modified as per the reviewer recommendations. GCC was defined in the main text where it first appears (Line 235 - 236)

Comment 8: Tables. Usually, the number of samples or observations is cited as “n”, more than “N”. In Table 2 is cited as “n”, and in other Tables as “N”. Please, be consistent.

Response 8: Thank you again. The text was modified as per the reviewer recommendations. Corrections were made in Line 769 and Table 1, 4, 6 and 7 to ‘n’

Comment 9: Lines 296-298: MAQ is defined in subheading. Usually, in subheadings it is not recommended to use abbreviations. However, if it should be defined the first time it appears in the main text (line 298)

Response 9: Agree and thank you. The text was modified as per the reviewer recommendations. In Line 183, first time MAQ appears and is defined as Meat Attachment Questionnaire as indicated.

Comment 10: Lines 421-423: This is not all true. It is true in North American Cultures, but in Europe in many countries the main meal is the midday meal and not the evening meal.

Response 10: Thank you for pointing this. The reviewer comments have been noted and modifications made accordingly (Lines 360 – 362).

Comment 11: Lines 506-509: nearly….fish” This rate is not according with those stablished in table 2. Please revise it.

Response 11: We thank you to the reviewer for the comment. Yes, agree and revised in lines 434 – 435.

Comment 12: References: Some sources are not cited according to MDPI guidelines. In example, “Meat Science” should be cited as “Meat Sci.”

Response 12: Thank you for pointing this. The text was modified as per the reviewer recommendations. Journal names abbreviated as per MDPI guidelines for reference no’s – 12, 19, 22, 23, 28, 29,54, 58, 63, 64, 65, 66, 68, 74, 75 highlighted in references.
